# Analysis of the Antibiotic Resistance Profiles in Methicillin-Sensitive *S. aureus* Pathotypes Isolated on a Commercial Rabbit Farm in Italy

**DOI:** 10.3390/antibiotics9100673

**Published:** 2020-10-05

**Authors:** Anna-Rita Attili, Alessandro Bellato, Patrizia Robino, Livio Galosi, Cristiano Papeschi, Giacomo Rossi, Eleonora Fileni, Martina Linardi, Vincenzo Cuteri, Francesco Chiesa, Patrizia Nebbia

**Affiliations:** 1School of Biosciences and Veterinary Medicine, University of Camerino, Via Circonvallazione 93/95, 62024 Matelica (MC), Italy; livio.galosi@unicam.it (L.G.); giacomo.rossi@unicam.it (G.R.); eleonora.fileni@studenti.unicam.it (E.F.); marti.linardi@gmail.com (M.L.); vincenzo.cuteri@unicam.it (V.C.); 2Department of Veterinary Sciences, University of Torino, Largo P. Braccini 2, 10095 Grugliasco (TO), Italy; alessandro.bellato@unito.it (A.B.); patrizia.robino@unito.it (P.R.); francesco.chiesa@unito.it (F.C.); patrizia.nebbia@unito.it (P.N.); 3Interdepartmental Animal Facility, University of Tuscia, Largo dell’Università snc, 01100 Viterbo (VT), Italy; papeschivet@gmail.com

**Keywords:** *Staphylococcus aureus*, rabbits, pathotype, antibiotic resistance profiles, spa type, Italy

## Abstract

The breeding of meat rabbits is an important sector in the livestock industry in Italy. The focus of this study was to describe the antibiotic resistance profile distribution among the Methicillin-sensitive *Staphylococcus aureus* isolated in a rabbit farm. From 400 animals of different ages and three farm workers, 96 randomly selected strains isolated from various anatomical sites and lesions were analysed. According to spa typing and the resistance profiles towards veterinary and human antibiotics, 26 pathotypes were identified. The highest resistance was observed against Tetracyclines (92.3%) and Macrolides (80.8%), while almost all were susceptible to Penicillins, according to the limited use of β-lactams on the farm. In total, 92.3% of pathotypes were multidrug resistant (MDRs). Two MDR pathotypes belonging to the t2802 spa type were isolated from both farmers and rabbits. Age categories harboured significantly different pathotypes (*p* = 0.019), while no association was found between pathotypes and lesions (*p* = 0.128) or sampling sites (*p* = 0.491). The antibiotic resistance was observed to increase with the time spent in the farm environment (age category). The selective pressure exerted by antibiotic use acted by giving advantage to more resistant strains rather than by lowering susceptibility to various drug categories within strains.

## 1. Introduction

The Italian rabbit industry annually produces about 24.5 million animals slaughtered for meat production and, in the European Union, takes third place after Spain and France. However, in recent years, rabbit breeding has seen a significant reduction in the number of commercial farms involved [1].

Rabbits are sensitive to many bacterial infections, such as respiratory and intestinal diseases, as well as skin infections. In general, farmers use various tools, such as biosecurity measures, good breeding conditions, alimentation, and behaviour. These protocols, when correctly implemented, give good results. Consequently, the use of antimicrobials has shown a steady decrease. Antibiotics are used as a therapy and include Fluoroquinolones, Trimethoprim-Sulfamethoxazole, Zinc Bacitracin, and Pleuromutilins, while other drug classes, such as Penicillins, Cephalosporins, and Lincosamides, need to be used with caution and should not be orally administered [2]. However, the use of antibiotics in livestock farming can increase antibiotic resistance prevalence, so it is also important to counteract this phenomenon under a One Health approach. Indeed, a low prevalence of antibiotic resistance in an intensive farm can be a sign of quality breeding [3,4].

*Staphylococcus aureus* is a zoonotic bacterium that occurs as both a commensal and opportunistic pathogen in many animals, including rabbits. It is associated with a wide variety of diseases, such as skin lesions, wound infections, mastitis, toxic shock syndromes, arthritis, endocarditis, osteomyelitis, and episodes of food poisoning [5,6,7]. In rabbits, this bacterium is an inhabitant of the skin, but it is also one on the main pathogens related to suppurative lesions [8]. For this reason, *S. aureus* could be used as indicator for the level of antimicrobial use in farms. Methicillin-resistant clones (MRSAs) are particularly monitored within the European community. In recent years, the presence of MRSAs in Italian rabbit breeding and the relationship between antibiotic use in farms and resistance have been evaluated [9,10]. High virulent strains (HV) were detected in Belgium, France, Greece, Italy, Portugal, Spain, and Hungary [8,11,12]. On the other hand, a recent study by Nemet et al. [12] provided evidence that low virulent (LV) strains can act as pathogens in rabbits.

Recently, a study showed various Methicillin-sensitive clones (MSSAs) in a rabbit farm, highlighting that some clonal strains (spa types) circulated in both animals and farm workers [13].

Although rabbit breeding is certainly a niche farm business, it has the potential risk for the spread of MDR, making it advisable to implement surveillance plans to control antibiotic resistance. In this research, following our previous study [13], the antibiotic resistance profile distribution among clones (spa types) isolated in an Italian rabbit farm was investigated. Moreover, to explore the evolution of antimicrobial resistance (AMR) in an intensively raised rabbit population, analyses and comparisons between the different pathotypes were carried out.

## 2. Results

### 2.1. Genetic Characterization

As previously reported by Attili et al. [13], in 96 randomly selected strains with low virulence and Methicillin-sensitive *S. aureus* (LV-MSSA), five different spa types were identified: t094, t491, t2036, t2802, and t605. The spa types most frequently found were t2802 (n = 51; 53.13%) and t491 (n = 37; 38.54%). Altogether, t094, t605, and t2036 accounted for 8.4%. Spa types were arranged into three spa-CCs: spa-CC267 (t2802), spa-CC084 (t094 and t491), and spa-CC012 (t2036). Due to its shortness, t605 could not be related to any clonal complex.

### 2.2. Antibiotic Susceptibility Testing

Antibiotic resistance was evaluated for each strain against 16 antibiotics belonging to 12 classes of antimicrobials. Table 1 reports the resistance prevalence for each of them.

All tested strains (n = 96) were susceptible to β-lactamase stable Penicillins (Cefoxitin), Lincosamides (Linezolid), Ansamycines (Rifampin), and Streptogramins (Quinupristin-Dalfopristin). The highest level of resistance was observed against Tetracyclines (n = 92, 95.8%). Out of the 96 strains, 90 (93.8%) were resistant to both of the Macrolides evaluated. Only three (3.1%) strains, obtained only from replacement rabbits, showed resistance to Penicillins. Susceptibility to Gentamicin and Tobramycin was highly correlated (r_tet_ = 99.0 ± 0.0), with 91 (94.8%) strains showing concordant responses. Therefore, these variables were shown to be dependent (*p* < 0.001). Moreover, resistance against Erythromycin and Clindamycin were dependent variables (*p* < 0.001), as they showed perfect concordance (r_tet_ = 1.0 ± 0.0). Among Glycopeptides, resistance against Teicoplanin and Vancomycin showed a weak negative correlation (r_tet_ = −0.2 ± 0.2), with only 27 (28.1%) strains showing concordant responses. Thus, the responses to the two antibiotics were shown to be independent from each other (*p* = 0.425). The overall resistance prevalence is shown in Figure 1.

Although an association between animal categories and resistance to antibiotics was observed only for Erythromycin and Clindamycin (Fisher’s exact *p* < 0.001), the resistance percentages varied among different spa types and age categories. Out of 96 strains, 92 (97.9%) were MDRs and were not associated with spa types (Fisher’s exact *p* = 0.161) but instead with age categories and adult and breeding rabbits (Fisher’s exact *p* = 0.010). Not-MDR strains were observed only in young and replacement rabbits, with an MDR prevalence of 66.7 ± 33.3% and 75.0 ± 25.0%.

### 2.3. Analysis of Resistance Profiles

Among the 96 strains, 23 unique different resistance profiles were found based on the susceptibility to each antibiotic (*a* to *w*) (Table 2). Out of the 23, 9 were associated with only one strain each, while the other 14 together accounted for 90.6% of the strains. The most frequent resistance profiles were *p* (n = 23, 24.0%) and *j* (n = 20, 20.8%). The six resistance profiles (*b*, *f*, *j*, *p*, *r*, and *s*) that were observed in five or more strains were not associated with the age category (χ^2^ = 12.447, *p* = 0.645).

Farmer strains shared the same resistance profiles with the rabbits: Profile *p* was isolated from 17 adult rabbits, three breeding rabbits, one young rabbit, and two farmers; *r* was drawn from two adult rabbits, two breeding rabbits, and one farmer. Most of the profiles had resistance to Tetracyclines (n = 21, 91.3%) and Macrolides (Erythromycin and Clindamycin: n = 18, 78.3%). Only two (8.7%) profiles (*t* and *w*) showed resistance to Penicillin G.

Out of the 23 profiles, 15 (65.2%) were resistant to Glycopeptides: One was susceptible to Teicoplanin but resistant to Vancomycin; 11 were resistant to Teicoplanin but susceptible to Vancomycin; and two were resistant to both antimicrobials. Only two (8.7%) resistance profiles (*i* and *s*) were not susceptible to Sulfamethoxazole-Trimethoprim, and 16 (69.6%) were resistant to Norfloxacin. About half of the profiles (n = 11, 47.8%) showed resistance to Aminoglycosides: One was resistant only to Gentamicin; two were resistant to Tobramycin but susceptible to Gentamicin; and eight were resistant to both.

Only six (26.1%) were resistant to Nitrofurantoin, and two (8.7%) profiles (*a* and *e*) showed resistance to less than three classes of antimicrobial agents, while 14 (60.9%) were resistant to 3–5 antimicrobial classes, and 7 (30.4%) were resistant to 6–8 classes. Therefore, all but *a* and *e* were MDRs, with an average of 5.1 ± 1.9 resistance against single antibiotics and slightly fewer (5.0 ± 1.4) resistance against classes of antimicrobials. Out of the 11 rabbits from which multiple strains were obtained, all were MDRs. Among them, eight showed discordant resistance profiles, while three showed concordant profiles. Out of the seven rabbits whose isolates were obtained from different sampling sites, five (71.4%) profiles were discordant. Out of the four rabbits whose isolates were drawn twice from the same sampling site, three (75.0%) had discordant profiles. No evidence was found showing that different sampling sites hosted different resistance profiles (Fisher’s exact *p* = 0.721).

### 2.4. Analysis of Pathotypes 

By associating the 23 resistance profiles to spa types, 26 pathotypes were identified (*A* to *Z*) (Table 2). Out of these 26, 10 were associated to only one strain each, while the other 16 together were found in 88.3% of the strains. The most represented pathotypes were *F* and *W*, which were isolated from 18 (18.8%) and 22 (22.9%) strains, respectively, with an estimated prevalence among rabbits of 18.1% (95CI: 10.3–25.9%) and 23.4% (95CI: 14.8–32.0%). Pathotypes observed more than once were associated with age categories (Fisher’s exact *p* = 0.019), but those that occurred at least five times (*D*, *E*¸ *F*, *W*, *Y*, and *Z*) were not (χ^2^ = 12.115, *p* = 0.670).

The two pathotypes isolated from farmers (*W* and *Y*) were also found in rabbits: *W* was isolated from 16 adult rabbits, three breeding rabbits, and one young rabbit; *Y* was drawn from two adult and two breeding rabbits. They were both spa type t2802 and resistant to six and seven classes of antimicrobials, respectively. Only two (7.7%) pathotypes (*B* and *C*), both drawn from replacement rabbits, showed resistance to Penicillin G.

Out of the 26 pathotypes, 17 (65.4%) were resistant to Glycopeptides; among them, one was susceptible to Teicoplanin but resistant to Vancomycin; 14 were resistant to Teicoplanin but susceptible to Vancomycin; and two were resistant to both antimicrobials. Only three (11.5%) pathotypes (*H*, *R*, and *Z*), drawn from fattening rabbits and breeders, were resistant to the Sulfamethoxazole-Trimethoprim combination, while 18 (69.2%) and 24 (92.3%), isolated from all four categories and farmers, showed resistance to Norfloxacin and Tetracycline, respectively. Half of the pathotypes (n = 13) obtained from all categories (except replacements) showed resistance to Aminoglycosides: One was resistant only to Gentamicin; two were resistant to Tobramycin but susceptible to Gentamicin; and 10 were resistant to both antibiotics. Only six (23.1%) pathotypes were resistant to Nitrofurantoin, while 21 (80.8%) drawn from all categories (except replacements) were found to be resistant to Macrolides (Erythromycin and Clindamycin). Only two (7.7%) pathotypes (*A* and *N*) were resistant to less than three classes of antimicrobial agents, while 15 (57.7%) were resistant to three to five antimicrobial classes, and nine (34.6%) to six to eight classes, with an overall 92.3% (n = 24) prevalence of MDRs.

Out of the 11 rabbits from which multiple strains were obtained, all reported MDRs. Among them, nine showed discordant pathotypes, while two showed concordant ones. Out of the seven rabbits whose isolates were obtained from different sampling sites, five (71.4%) pathotypes were discordant. All four rabbits whose isolates were drawn twice from the same site had discordant pathotypes. Consistently, no evidence showed that different sampling sites harboured different pathotypes (Fisher’s exact *p* = 0.491).

The complexity of data, due to the high number of different pathotypes, and the scarcity of many of them in the population did not allow us to obtain further information on the distribution of the lineages among age categories in the rabbit population.

### 2.5. Clusterization and Factor Analysis 

Since some pathotypes showed similar resistance profiles, factor analysis (FA) was used to reduce the complexity of the database and group strains based on the similarity of their phenotypic and genotypic features. As Erythromycin and Clindamycin were completely correlated, they were aggregated in FA to avoid a singularity of the correlation matrix. The sample was adequate to perform the factor analysis (Kaiser–Meyer–Olkin measure = 0.686). FA produced 11 factors, of which three were retained because they could explain 98.7% of the data variability. The variables that contributed the most to variability were Aminoglycosides (Gentamicin 90.4%, Tobramycin 94.7%), spa type (89.0%), Norfloxacin (72.1%) for the first factor; Penicillin G (79.4%) and Erythromycin (72.8%) for the second one; and Tetracycline (74.0%) for the third factor (Figure 2).

Other variables were less relevant to defining data variability. Since the first three factors were sufficiently informative to cover the variability of the whole dataset, they were used to produce the similarity matrix. Based on the matrix and a cut-off value of 0.0, five different clusters of strains were generated. Spa types were partly grouped together, but they were not completely separated from each other, with t491 showing high variability. Pathotypes were grouped together in a three-dimensional space as follows: Cluster 1: *D*, *F*, *J*, *I*, *M*; Cluster 2: *A*, *B*, *C*, *E*, *G*, *N*, *X*; cluster 3: *O*, *T*, *U*, *V*; cluster 4: *H*, *K*, *L*, *S*; and cluster 5: *P*, *Q*, *R*, *W*, *Y*, *Z*. The score-plot in Figure 3 shows the spatial distribution of the clusters and pathotypes in a two-dimensional space.

### 2.6. Analysis of Resistance Number 

The resistance varied significantly among spa types (Friedman’s test *p* = 0.002), ranging from 3.5 ± 1.0 in t094 to 5.9 ± 1.2 in t2802, but did not differ among age categories (Friedman’s test *p* = 0.815), ranging from 3.5 ± 0.5 in replacement rabbits to 5.5 ± 0.3 in breeding rabbits. However, an increasing resistance trend was appreciable among the age categories of rabbits and the strains obtained from farmers, which showed the highest average resistance value (6.3 ± 0.6) (Figure 4). 

According to the age categories, the resistance increased significantly with the time spent in the farm environment (Table 3).

Since spa types were associated with both age categories and time on the farm (Fisher’s exact *p* < 0.001), as well as resistance profiles the categories needed to be added to the regression model to evaluate the “time on farm” effect on the resistance count. Due to including the fewest resistance, t2036 was set as the reference group. t605 and t2802 proved to be significantly related to an increase in resistance types. After the inclusion of spa types in the model, the effect of the time on farm was no longer significant (*p* > 0.2). Results and estimates for spa types are reported in Table 4.

Figure 5 shows that the significantly more resistant spa types (t605 and t2802, n = 53) increased with the time spent on the farm, while the other spa types (n = 43) reduced (Figure 5).

Although the time-on-farm effect vanished when spa types were considered, this effect could still affect the prevalence of different pathotypes. Therefore, to consider the full complexity of the data, an evaluation of the number of resistance types was performed in relation to pathotype clusters and the time-on-farm variable. The time spent in the farm environment proved not to be significant overall (χ^2^ = 4.09, *p* = 0.129), except between the first (<60 days) and the third (≥240 days) categories (χ^2^ = 3.93, *p* = 0.047).

Clusters had significantly different counts of resistance overall (χ^2^ = 519.17, *p* < 0.0001). Compared to cluster 1, cluster 2 did not have significantly more resistance (χ^2^ = 0.68, *p* = 0.411); cluster 3 had significantly more resistance than cluster 1 (χ^2^ = 4.63, *p* < 0.032) but not more than cluster 2 (χ^2^ = 3.02, *p* < 0.082); clusters 4 and 5 had significantly more resistance than clusters 1 and 2 (*p* < 0.001) but not more than cluster 3 (χ^2^ = 0.5, *p* < 0.480; χ^2^ = 2.58, *p* < 0.108, respectively). The increasing trend acted on two different levels: Macroscopic variations were appreciable between clusters, while there were minor intra-cluster differences among the time-on-farm categories. Table 5 outlines the estimated counts of resistance by the clusters and time on farm of the categories.

Figure 6 shows the predicted counts of resistance for each cluster, according to the on-farm time variable.

The chart shows that there is an effect of clusters that is also dependent from the time on farm, which had an additive effect between the first and the third category. There was also a break in the trend between the cluster 2 and 3 values.

## 3. Discussion

The high prevalence of multidrug-resistant strains observed in this study (97.9%) agrees with what was previously described in intensively raised rabbits [14]. The highest resistance observed in this study, particularly against Tetracycline (95.8%), Macrolides (93.8%), and Quinolones (63.5%), is also similar to that observed against the same antibiotics in a recent study conducted in Spain and Portugal [14]. An unexpectedly high resistance was observed against Glycopeptides (77.1%), whose use is restricted to human medicine. This result suggests the possibility of human–animal transmission, although this study was not designed to test this factor. Moreno-Grua et al. [14] observed a 12.5% prevalence of Methicillin-resistant strains in the Iberian Peninsula, isolated from 22 out of 89 farms. We did not find any Methicillin-resistant strain, which is almost in agreement with the only MRSA case published on rabbits in Italy [9] that reported only one in 40 intensive rabbit farms having MRSA.

Due to its simplicity and affordability, spa typing has been described as a useful tool to survey the circulation of pathogenic strains in farm environments and among different farms and countries [15]. The clonal phylogenetic structure of *S. aureus* allows one to trace back the origin of the bacterial strains and map the lineages’ evolution in a population, although it cannot fulfil all the needs of epidemiological surveillance of antimicrobial resistance. In this study, a representative sample of a relatively closed population of rabbits was chosen to investigate the prevalence of antibiotic-resistant Methicillin-sensitive *S. aureus* strains. Five spa types were found circulating among the animals, unevenly colonizing different body sites and age categories [15]. It was also observed that some spa types were more strongly related to some age categories, such as t094 in replacement rabbits. Since these animals were partly bought from outside the farm, it could be presumed that *S. aureus* strains were introduced along with the animals, while also bringing in new resistance genes. However, the overall group of newly introduced rabbits was very reduced (approximately 2.6%). Further, although these rabbits might host different spa types, those strains should have an evolutionary advantage in establishing themselves in a population already colonized by *S. aureus*. One of the main forces acting as selective pressure on bacteria in a farm environment is the usage of antibiotics, which act not only on pathogens but also on commensal flora that are being actively selected towards an increase in resistance [16,17]. To survey the evolution of resistant strains, spa typing showed its limit, as it was unable to differentiate strains at the level of detail needed. These details lay in the genomes of bacteria as resistance genes. However, for the purpose of this study, phenotypic manifestation was used to maintain a pragmatic approach since most laboratories do not have the resources to perform multiple types of molecular testing to identify resistance genes. Since phenotypic resistance patterns likely reflect underlying genotypic differences, susceptibility to each antibiotic could be considered a distinctive tract useful to define lineages circulating in the farm environment, along with spa types. 

Out of the 8192 possible arrangements, only 23 different resistance profiles were found. Therefore, it is reasonable to expect that they are related to some other features of the strains or even to each other. The pathotype identification hinted at an association between spa types and resistance profiles, but such pathotypes were too dispersed to more deeply investigate. The implementation of factor analysis on the phenotypic data was intended to enhance the readability of the database, thereby improving our ability to understand how similar these pathotypes are. The results showed that spa typing does not describe all variability but remains a powerful tool to clarify clonal evolution mechanisms; further, the clusters of pathotypes generated by FA were associated with age categories.

Our objective was to explore how the selective pressure exerted by antimicrobial treatments affects the amount of resistance in *S. aureus*. Therefore, we assumed that the time spent in the farm environment would be a proxy for the quantity of exposure to antibiotics and that the farm environment would be a source of colonization for most resistant strains of *S. aureus*, which are considered to have the best survival chances. To test these assumptions, age categories were redefined while considering the physiology of rabbits and intensive-breeding times. By simply evaluating the frequencies of the most resistant spa types (t605 and t2802), it was observable that their prevalence increased with exposure to the farm environment and that the time on farm had a significant effect on the amount of resistance. A causal relationship was also supported by the dose–effect trend and prior knowledge of biological mechanisms. However, the time on farm was no longer able to statistically significantly explain this increase when the clusters of pathotypes were considered, although the trend remained evident. These findings confirm that the increase in resistance was due to antibiotic usage and the selective pressure exerted, while also suggesting another question: With an increase in time spent on the farm, do the resistant strains increase or does the resistance in strains increase? This sounds trivial, but, based on our findings, the former option seems more probable than the latter since the relationship between strains and resistance can completely hide the relationship between age and resistance. On the other hand, it is reasonable to assume that both mechanisms play a role in AMR, but they might proceed at different speeds. 

More resistant strains are considered to have the best survival chances; therefore, their prevalence increased as they filled the space left empty by those that did not survive the antibiotic treatment. This finding indicates that rabbits who live for a long time in the environment play a key role in maintaining resistant strains and spreading them to newly introduced and new-born individuals. Furthermore, the role of farmers should not be underestimated, as farmers survive more often than animals and continue to spread resistant strains to newly introduced rabbits. The usefulness of the sanitary vacuum remains debated because it only applies to animals and the environment, but farmers could also act as a source of resistant strains.

## 4. Materials and Methods 

### 4.1. Sampling and S. aureus Characterization 

Sampling and bacteriological analysis (identification and molecular characterization of *S. aureus* isolates) were described in detail in our previous article [13] and are briefly summarized below.

Out of a total of 2066 samples taken from 400 rabbits of different ages reared in a commercial medium-sized intensive breeding farm in central Italy, and from farmworkers, 592 Methicillin-sensitive *S. aureus* strains (n = 552 from healthy skin and n = 40 from lesions) were isolated. A sample of 96 *S. aureus* was randomly selected to perform antimicrobial susceptibility tests (ASTs) and molecular analyses. The sample was so distributed: 72 strains representing the four breeding categories (n = 2, 2.8% of *S. aureus* infection in youngsters; n = 52, 72.2% in adults; n = 14, 19.4% in breeding rabbits; n = 4, 5.5% in replacement rabbits) were taken from healthy skin, 21 strains were isolated from lesions, and three from the healthy skin of the nose and hands of the farm workers, who gave their informed consent. Animals treated with antibiotics within 30 days were excluded from the study; the farm’s veterinarian reported data about the antibiotic treatment in the previous three years: Enrofloxacin, Sulfamethoxazole-Trimethoprim, Zinc Bacitracin, and Valnemulin were used to treat bacterial infections as need.

### 4.2. Antibiotic Susceptibility Testing

To determine the antibiotic profiles, 96 randomly selected *S. aureus*, *mecA,* and *mecC* negative (MSSA strains) were screened, and antibiotic susceptibility testing for 16 antimicrobial agents was performed according to the Clinical and Laboratory Standards Institute guidelines [18,19].

*S. aureus* subsp. *aureus* ATTC^®^ 43300TM was used as a quality control strain for the zone diameter determinations and MIC evaluation. The standard disk diffusion method (Kirby-Bauer test) was used to test Penicillin G (P, 1 IU), Cefoxitin (FOX, 30 µg), Sulfamethoxazole-Trimethoprim (SXT, 25 µg), Norfloxacin (NOR, 10 µg), Tetracycline (TE, 30 µg), Gentamicin (GM, 10 µg), Tobramycin (TOB, 10 µg), Erythromycin (E, 15 µg), Clindamycin (CD, 2 µg), Rifampin (RIF, 5 µg), Nitrofurantoin (NT, 100 µg), Linezolid (LIN, 10 µg), and Quinupristin-Dalfopristin (Q-D, 15 µg). As recommended by CLSI (2018), Penicillin 1 IU was used to test the isolates against Penicillins and as a marker for penicillinase-labile beta-lactams, while Cefoxitin was used to test the strains against the penicillinase-stable ones (phenotypic resistance to methicillin). The E-test method was used to determine the MICs against Amoxicillin-Clavulanate, Vancomycin, and Teicoplanin, as described by the manufacturer (MIC Test Strip, Liofilchem, Roseto degli Abruzzi, Italy). Isolates that exhibited intermediate sensitivity were considered resistant to the antibiotic.

Some drugs were chosen for their importance in treating *S. aureus* infections in animals and others due to their ability to provide diversity in the representation of different antimicrobial agent classes. As Linezolid, Quinupristin-Dalfopristin, Vancomycin, and Teicoplanin are among the antibiotics that have become life-saving treatments for human patients suffering from different kinds of infections caused by multidrug-resistant bacteria, it was in our interest to assess sensitivity to these antimicrobials. Profiles of antibiotic resistance were investigated singularly and in association with clones (spa types) to define the unique combinations of phenotypical and genotypical characteristics as pathotypes. Alphabetical codes were used to identify resistance profiles (lowercase) and pathotypes (uppercase).

Finally, the pathotypes were grouped together in clusters based on spa types and resistance patterns.

### 4.3. Statistical Analysis

Many categorical variables were used to perform the statistical analyses, some related to the animals (rabbit’s identification code, age category, site of sampling) and others to the bacteria (spa type). Resistance to antibiotics was coded as 1 if the strain was intermediate or resistant and 0 otherwise. From the previous variables, the other variables were derived: MDR strains were defined as those having resistance against at least three different antimicrobial classes [20]; the count of resistance was obtained by adding together all resistance types against single antibiotics. 

According to a farmer’s statement, on average, 30% of replacement rabbits were introduced from outside the farm at an adult age to replace the breeding rabbits (personal communication). For this reason, these rabbits could be conveniently associated with the young rabbits due to their short average time spent in the farm environment. On the other hand, breeding rabbits and farmers have spent a long time in the farm environment, hinting at a possible reunion of these two categories. Therefore, the age category variable was recorded as the time on farm as follows: <60 days, grouping together young rabbits that were 40–45 days of age with replacement rabbits; 60–80 days, for the age of adult rabbits that had lived on the farm since birth; and ≥240 days for grouping breeding rabbits whose reproductive activity started at least at 8 months of age with farmers. The time on farm analysis was performed as an exploratory survey since it was not defined from the beginning, but the need for it arose over the course of the analyses.

Data analysis proceeded as follows: (1) analysis of individual strains and the overall prevalence of resistant ones; (2) analysis of the resistance profiles and associations with age category and spa type and the prevalence of MDRs; (3) analysis of the pathotypes and their association with age categories; (4) factor analysis and hierarchical clustering of pathotypes; and (5) analysis of the count of resistance and its association with other variables.

To investigate the prevalence of resistant strains *versus* antibiotics, as well as antimicrobial classes, an exploratory analysis was performed to calculate the mean, standard deviation, and standard error of the mean. Contingency tables were produced to investigate the relationships between resistance and spa types, age categories, and sampling sites using chi-square and Fisher’s exact tests to assess the associations. The correlation between resistance to antibiotics belonging to the same class was evaluated with the tetrachoric correlation coefficient, r_tet_ [21]. Resistance profiles were obtained by collapsing the database to unique combinations of resistance to antibiotics and identified by lowercase letters from “*a*” to “*w*”. The resistance profiles’ prevalence and association with the age categories and spa types were estimated via the same methods described for individual strains. Pathotypes were defined as unique combinations of resistance profiles and spa types assigned uppercase letters from “*A*” to “*Z*”. The prevalence of pathotypes resistant to every antibiotic and the association of this resistance with age categories were also investigated.

To improve data readability, enhance information quality, and avoid tautologies, a factor analysis (FA [22]) was performed on a set of 11 variables (spa types and resistance to antibiotics). FA produced some factors (vectors of eigenvalues) able to cover the variability of the related variables. The Kaiser–Meyer–Olkin measure was calculated to assess the usefulness of FA. The relationship among the former variables and the main factors was evaluated graphically by a loading plot and quantitatively by the explained cumulative variance. Using a graphical evaluation of the scree plot, some factors were chosen and then used for hierarchical clustering by applying Ward’s criterion to the similarity matrix [23]. The obtained clusters resembled the groups of pathotypes circulating on the farm environment. 

An analysis of variance to assess the resistance among age categories and spa types was performed using Friedman’s test. The trend of the amount of resistance was evaluated using Poisson’s regression models considering either the age category and spa types or the time on farm and clusters. The robust standard error was calculated.

Data were recorded on a Microsoft Excel spreadsheet, which was also used to build a database. Every subsequent analysis was performed using STATA 15.0 (Stata Corp. LLC, College Station, TX, USA).

## 5. Conclusions

This study showed that 92.3% of pathotypes were MDRs and associated with age categories and adult and breeding rabbits. The highest resistance was observed against the classes of Tetracyclines, Macrolides, and Glycopeptides in both animal and human *S. aureus* strains. According to the analysis of resistance, 23 unique different profiles were obtained, with farmer strains that shared the same resistance profiles with rabbits. In particular, when the 23 resistance profiles were associated with the spa types, 26 pathotypes were identified. Two pathotypes isolated from farmers were also found in rabbits, which were both spa type t2802 and resistant to six and seven class of antimicrobials, respectively, confirming the zoonotic risk that could occur in rabbit farms.

As Nemet et al. [14] observed, clonal types do not always correspond to spa types, through which only part of the bacterial genome is sequenced and evaluated. However, information on lineages circulating in the herd obtained by cluster analysis on antimicrobial resistance profiles could help us understand how the selective pressure exerted by antimicrobial treatment acts on *S. aureus* strains and thus provide useful management indications for antibiotic use.

## Figures and Tables

**Figure 1 antibiotics-09-00673-f001:**
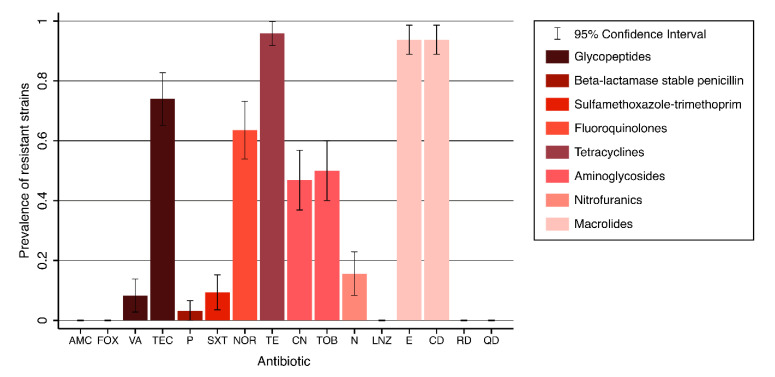
Prevalence of resistance against each antibiotic, with error bars showing 95% confidence intervals (AMC: Amoxicillin-Clavulanate; FOX: Cefoxitin; VA: Vancomycin; TEC: Teicoplanin; P: Penicillin G; SXT: Sulfamethoxazole-Trimethoprim; NOR: Norfloxacin; TE: Tetracycline; CN: Gentamicin; TOB: Tobramycin; N: Nitrofurantoin; LNZ: Linezolid; E: Erythromycin; CD: Clindamycin; RD: Rifampin; QD: Quinupristin-Dalfopristin).

**Figure 2 antibiotics-09-00673-f002:**
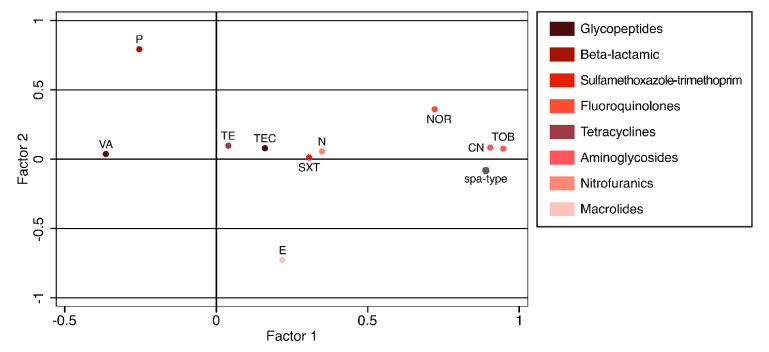
Loading plot showing the impact of each variable on the first two factors of the factor analysis. Legend: VA = Vancomycin; P = Penicillin G; TE = Tetracycline; TEC = Teicoplanin; E = Erythromycin; SXT = Sulfamethoxazole-Trimethoprim; N = Neomycin; NOR = Norfloxacin; CN = Gentamicin; TOB = Tobramycin.

**Figure 3 antibiotics-09-00673-f003:**
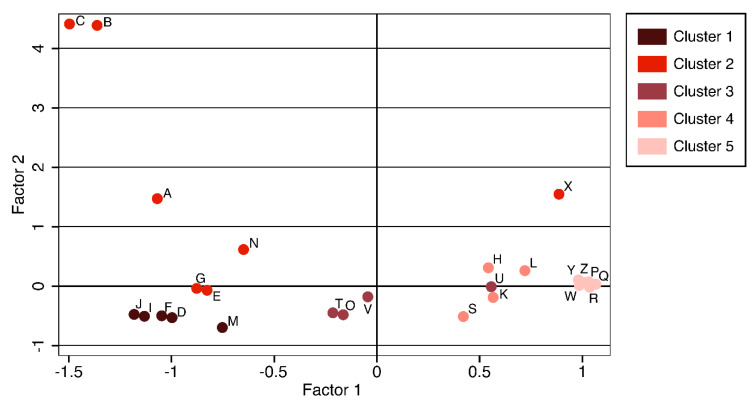
Scores of the pathotypes and their positions relative to the first two factors. Clusters are identified by different colours.

**Figure 4 antibiotics-09-00673-f004:**
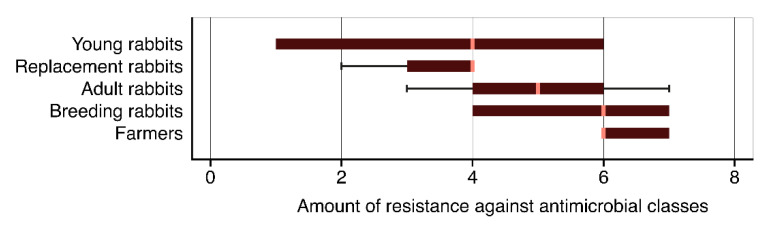
Resistance against antimicrobial classes for rabbits (age categories) and farmers. Legend: Error bars = Inter-quartile range.

**Figure 5 antibiotics-09-00673-f005:**
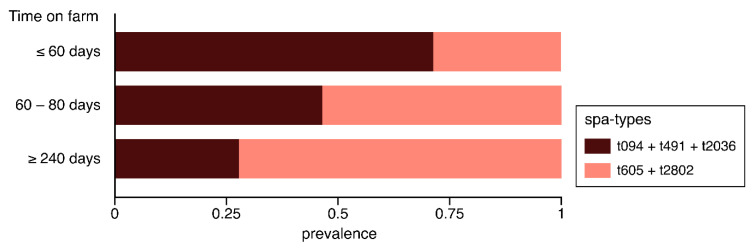
Prevalence of significantly more resistant spa types (t605 and t2802) against the less resistant ones for each in-farm time category.

**Figure 6 antibiotics-09-00673-f006:**
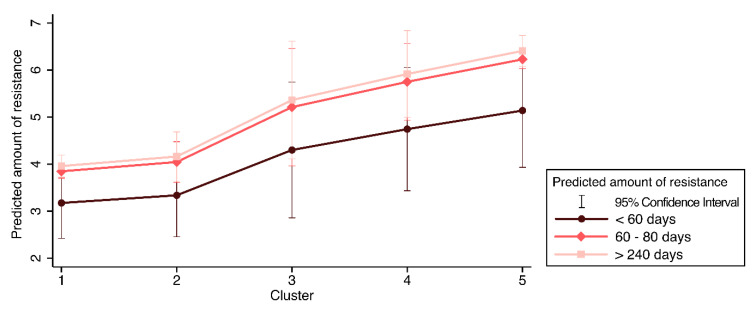
Predicted counts of resistance estimated considering the clusters and time on farm of the animal categories. Legend: error bars = 95% Confidence Interval of the predicted mean.

**Table 1 antibiotics-09-00673-t001:** Frequency and resistance prevalence against the 12 antimicrobial classes and 16 antibiotics.

Antimicrobial Class	Freq.	% Prevalence (Mean ± s.e.)	Antibiotic	Freq.	% Prevalence (Mean ± s.e.)
Aminoglycosides	49	51.0 ± 0.5	Gentamicin	45	46.9 ± 0.5
			Tobramycin	48	50.0 ± 0.5
Ansamycins	0	0.0 ± 0.0	Rifampin	0	0.0 ± 0.0
β-lactamase labile Penicillins	3	3.1 ± 0.2	Penicillin G	3	3.1 ± 0.2
β-lactamase stable Penicillins	0	0.0 ± 0.0	Cefoxitin	0	0.0 ± 0.0
			Amoxicillin-Clavulanate	0	0.0 ± 0.0
Folate Pathway Inhibitors	9	9.4 ± 0.3	Sulfamethoxazole-Trimethoprim	9	9.4 ± 0.3
Fluoroquinolones	61	63.5 ± 0.5	Norfloxacin	61	63.5 ± 0.5
Glycopeptides	74	77.1 ± 0.4	Teicoplanin	71	74.0 ± 0.5
			Vancomycin	8	8.3 ± 0.3
Macrolides	90	93.8 ± 0.3	Clindamycin	90	93.8 ± 0.3
			Erithromycin	90	93.8 ± 0.3
Nitrofuranics	15	15.6 ± 0.4	Nitrofurantoin	15	15.6 ± 0.4
Lincosamides	0	0.0 ± 0.0	Linezolid	0	0.0 ± 0.0
Streptogramins	0	0.0 ± 0.0	Quinpristin-Dalfopristin	0	0.0 ± 0.0
Tetracyclines	92	95.8 ± 0.2	Tetracycline	92	95.8 ± 0.2

Legend: s.e. = standard error.

**Table 2 antibiotics-09-00673-t002:** Spa types associated to resistance profiles and pathotypes and their relative frequencies. All strains were susceptible to Amoxicillin-Clavulanate, Cefoxitin, Linezolid, Rifampin, and Quinupristin-Dalfopristin, so these antibiotics are not shown.

Spa Type	Freq. (%)	Resistance Profile	Pathotype	Freq. (%)	VA	TEC	P	SXT	NOR	TE	CN	TOB	N	E	CD
t094	4 (4.2)	*e*	*A*	1 (1.0)	S	S	S	S	R	R	S	S	S	S	S
		*t*	*B*	2 (2.1)	S	R	R	S	R	R	S	S	S	S	S
		*w*	*C*	1 (1.0)	R	R	R	S	R	R	S	S	S	S	S
t491	37 (38.5)	*b*	*D*	5 (5.2)	S	S	S	S	S	R	S	S	S	R	R
		*f*	*E*	5 (5.2)	S	S	S	S	R	R	S	S	S	R	R
		*j*	*F*	18 (18.8)	S	R	S	S	S	R	S	S	S	R	R
		*m*	*G*	1 (1.0)	S	R	S	S	R	R	S	S	S	R	R
		*s*	*H*	1 (1.0)	S	R	S	R	R	R	R	R	S	R	R
		*u*	*I*	3 (3.1)	R	S	S	S	S	R	S	S	S	R	R
		*v*	*J*	4 (4.2)	R	R	S	S	S	R	S	S	S	R	R
t605	2 (2.1)	*c*	*K*	1 (1.0)	S	S	S	S	S	R	R	R	R	R	R
		*p*	*L*	1 (1.0)	S	R	S	S	R	R	R	R	S	R	R
t2036	2 (2.1)	*j*	*M*	2 (2.1)	S	R	S	S	S	R	S	S	S	R	R
t2802	51 (53.1)	*a*	*N*	1 (1.0)	S	S	S	S	S	R	S	S	S	S	S
		*d*	*O*	2 (2.1)	S	S	S	S	R	S	S	S	S	R	R
		*g*	*P*	1 (1.0)	S	S	S	S	R	R	R	R	S	R	R
		*h*	*Q*	4 (4.2)	S	S	S	S	R	R	R	R	R	R	R
		*i*	*R*	2 (2.1)	S	S	S	R	R	R	R	R	S	R	R
		*k*	*S*	1 (1.0)	S	R	S	S	S	R	S	R	S	R	R
		*l*	*T*	2 (2.1)	S	R	S	S	R	S	S	S	S	R	R
		*n*	*U*	3 (3.1)	S	R	S	S	R	R	S	R	R	R	R
		*o*	*V*	1 (1.0)	S	R	S	S	R	R	R	S	R	R	R
		*p*	*W*	22 (22.9)	S	R	S	S	R	R	R	R	S	R	R
		*q*	*X*	1 (1.0)	S	R	S	S	R	R	R	R	R	S	S
		*r*	*Y*	5 (5.2)	S	R	S	S	R	R	R	R	R	R	R
		*s*	*Z*	6 (6.3)	S	R	S	R	R	R	R	R	S	R	R

VA: Vancomycin; TEC: Teicoplanin; P: Penicillin G; SXT: Sulfamethoxazole-Trimethoprim; NOR: Norfloxacin; TE: Tetracycline; CN: Gentamicin; TOB: Tobramycin; N: Nitrofurantoin; E: Erithromycin; CD: Clindamycin.

**Table 3 antibiotics-09-00673-t003:** Observed and estimated number of class resistance by age category and in-farm time recorded groups. Risk ratios for the increase of youngest one.

Age Category	Resistance (Mean ± s.e.)	Time on Farm (Days)	Freq. (%)	Estimated Resistance (No.)	Estimated 95% C.I.	RR	*p*-Value
Replacement rabbits	3.5 ± 0.5	<60	7 (7.29)	3.6	2.5–4.7	1 (ref.)	-
Young rabbits	3.7 ± 1.5						
Adult rabbits	5.0 ± 0.2	60–80	71 (74.0)	5.0	4.7–5.3	1.4	0.035
Breeding rabbits	5.5 ± 0.3	≥240	18 (18.8)	5.6	5.1–6.2	1.6	0.007
Farmers	6.3 ± 0.3						

Legend: s.e. = standard error; C.I. = Confidence Interval; RR = Relative Risk.

**Table 4 antibiotics-09-00673-t004:** Observed and estimated resistance types for each spa type. Relative risks of gaining one more type(s) of resistance in each group compared to t2036.

Spa Type	Resistance (Mean ± s.e.)	Estimated Resistance (No.)	Estimated 95% C.I.	RR	*p*-Value
t094	3.5 ± 0.5	5.0	1.5–8.5	1.3	0.464
t491	4.0 ± 0.1	3.9	3.7–4.2	1.0	0.372
t605	5.5 ± 0.5	5.4	4.7–6.1	1.4	<0.001
t2036	4.0 ± 0.0	3.8	3.5–4.1	1 (ref.)	-
t2802	5.9 ± 0.2	5.9	5.5–6.2	1.5	<0.001

Legend: s.e. = standard error; C.I. = Confidential Interval; RR = Relative Risk.

**Table 5 antibiotics-09-00673-t005:** Predicted counts of resistance by the clusters and time on farm of the categories.

Cluster	Time on Farm (Days)	Estimated Resistance (No.)	Estimated 95% C.I.
1	<60	3.2	2.4–3.9
	60–80	3.9	3.7–4.0
	≥240	4.0	3.7–4.25
2	<60	3.3	2.5–4.2
	60–80	4.0	3.6–4.5
	≥240	4.2	3.6–4.7
3	<60	4.3	2.9–5.7
	60–80	5.2	4.0–6.5
	≥240	5.4	4.1–6.6
4	<60	4.8	3.4–6.1
	60–80	5.8	4.9–6.6
	≥240	5.9	5.0–6.8
5	<60	5.1	3.9–6.3
	60–80	6.2	6.0–6.4
	≥240	6.4	6.1–6.7

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
