# Peer review of "Analysis of the Antibiotic Resistance Profiles in Methicillin-Sensitive S. aureus Pathotypes Isolated on a Commercial Rabbit Farm in Italy"

_antibiotics, 2020, doi:10.3390/antibiotics9100673_

Round 1
Reviewer 1 Report
The research group's research will present the results on drug resistance of strains silenced from rabbits and workers of the breeding form. The work's subject raises an essential aspect of the transmission of multi-drug resistant strains between animals and humans, which may be a significant problem. However, the above work needs improvement.
1. The abstract is unclear, and there are errors in it, e.g., line 27 is missing a comma.
2. The introduction lacks a few references that the authors could refer to.
3. There is also no information about the number of samples taken, whether they were always taken only from the lesions.
4. The withdrawal period of 30 days from antibiotics seems to be too short, in my opinion.
5. Some sentences appear very long so that the recipient does not always know what the authors mean.
6. Out of 19 references, as many as 10 are older than 10 years, which somewhat disregards the work as a less current topic.
7. Missing Table 5. and Figure 6. is captioned at the top instead of at the bottom.
8. There are errors, for example. In line 84, having 'β-lattamase" instead of" β-lactamase ".
9. The names of antibiotics are also not uniformly written, sometimes they are in capital letters and sometimes in lower case letters.
In general, the subject of the work is exciting, but it does not touch on innovative aspects, there are many works in a similar field published in recent years. The authors do not entirely refer to the hypothesis/assumptions of the work in their conclusions.
Author Response
ANTIBIOTICS - 901983, REVISION NOTE
Reply to the comments on manuscript “Analysis of the Antibiotic Resistance Profiles in Methicillin-Sensitive S. aureus Pathotypes Isolated on a Commercial Rabbit Farm in Italy “
Author’s Response to the Reviewer’ Comments
Reviewer #1
The authors thank the reviewers for their useful comments and suggestions to the manuscript.
All comments have been taken in due consideration and the corrections added to the original paper. In the manuscript all corrections are highlighted in yellow color.
The present version of the manuscript has been checked by the MDPI’s English editing service.
We hope that the manuscript is now suitable for publication.
Thank you again for your consideration
Sincerely,
The authors
The abstract is unclear and there are errors in it, e.g., line 27 is missing a comma.
Thank you for the observation, we tried to better clarify the results obtained, while maintaining the maximum number of words allowed (lines: 22, 24-33). Errors have been corrected and highlighted in yellow color.
The introduction lacks a few references that the authors could refer to.
The authors followed the suggestion and they have updated the references (lines: 59-61).
There is also no information about the number of samples taken, whether they were always taken only from the lesions.
As suggest, we have added the number of samples taken (lines: 72, 349-351).
The withdrawal period of 30 days from antibiotics seems to be too short, in my opinion.
The comment is true and relevant, but in rabbit farm the breeding phases are very short. A full cycle lasts 90 days. We have therefore deemed a period of non-treatment of at least 30 days to be sufficient.
Some sentences appear very long so that the recipient does not always know what the authors mean.
The authors corrected all the manuscript and shortened some sentences, also in agreement with the English MDPI’s editing service.
Out of 19 references, as many as 10 are older than 10 years, which somewhat disregards the work as a less current topic.
The authors updated the bibliography (lines: 465, 469, 481, 487).
Missing Table 5 and Figure 6 is captioned at the top instead of at the bottom.
The authors corrected and made visible the table 5 that resulted in the text but sometimes hidden. In case, it appears by clicking on the triangular icon, on the left.
There are errors, for example: in line 84, having 'β-lattamase" instead of" β-lactamase ".
We are sorry for the incorrect editing. As suggest, we have revised the text. Corrections have been made and are highlighted in yellow.
The names of antibiotics are also not uniformly written, sometimes they are in capital letters and sometimes in lower case letters.
You are right, we used capital letters for names identifying the antibiotics’ class and lowercase letters for the names of the antibiotics. Now we have rewritten all names in capital letter.
.. The authors do not entirely refer to the hypothesis/assumptions of the work in their conclusions. Following this recommendation, we resumed the link between aims and our results (lines 263-273).
Moreover, regarding antibiotic resistance results we have expanded this point at the beginning of the discussion.
A chapter ‘Conclusions’ was added (lines 332-344).
Reviewer 2 Report
Dear authors
The study titled ‘Analysis of Antibiotic Resistance Profiles in Methicillin-Sensitive S. aureus Pathotypes Isolated in a Commercial Rabbit Farm in Italy’ represents the completion of a previously published research.
Although the study is limited to a single farm, this manuscript is reasonably well presented and it contributes to the acquisition of relevant epidemiological knowledges on the circulation of different S. aureus pathotypes in a rabbit farm reared for meat production in Italy. Although all S. aureus strains were sensitive to methicillin, the study of resistance profiles to 12 classes of antibiotics used in veterinary medicine and/or human medicine, contributes to a greater understanding about the real different S. aureus pathotypes, including zoonotic ones, belonging to the same spa-type.
The study is methodically well described. The statistical analysis was carried out correctly by evaluating not only the observed resistance frequencies but also the association of the resistance profiles with different variables,
resulting in the study and identification of real different genotypic and phenotypic strains: pathotypes. Clusterization and factor analysis were also performed.
Interesting results were observed for human antibiotics, demonstrating the zoonotic role and the effect of the spread and circulation of pathotypes from animals to humans and from humans to animals.
All the comments and suggestions I have made are of a minor nature and I would be happy for the manuscript to be accepted for publication once these have been addressed.
In particular:
Line 86: please replace in italics ‘versus’.
Line 101: please, insert the error bars explanation in legend.
Line 105: please clarify which age animal category showed the highest percentage of MDR resistance
Line 110: please, reformulated more clearly for the reader this proposition ‘Profiles with a frequency equal or higher than 5 …’
I Hope my reviewing could help you to improve your work.
Regards
Author Response
ANTIBIOTICS - 901983, REVISION NOTE
Reply to the comments on manuscript “Analysis of the Antibiotic Resistance Profiles in Methicillin-Sensitive S. aureus Pathotypes Isolated on a Commercial Rabbit Farm in Italy “
Author’s Response to the Reviewer’ Comments
Reviewer #2
The authors thank the reviewers for their useful comments and suggestions to the manuscript.
All comments have been taken in due consideration and the corrections added to the original paper. In the manuscript all corrections are highlighted in yellow colour.
The present version of the manuscript has been checked by the MDPI’s English editing service.
We hope that the manuscript is now suitable for publication.
Thank you again for your consideration
Sincerely,
The authors
All the comments and suggestions I have made are of a minor nature and I would be happy for the manuscript to be accepted for publication once these have been addressed. In particular:
Line 86: please replace in italics ‘versus’. The MDPI’s English editing service corrected the word ‘versus’ with ‘against’.
Line 101: please, insert the error bars explanation in legend. The correction has been made.
Line 105: please clarify which age animal category showed the highest percentage of MDR resistance. As suggested, we have better specified (lines 107-109).
Line 110: please, reformulated more clearly for the reader this proposition ‘Profiles with a frequency equal or higher than 5 …’ The sentence has revised as suggest (lines 114-115).
Reviewer 3 Report
It has been a great honor, as well as a pleasantly challenging activity, to review the article entitled ”Analysis of Antibiotic Resistance Profiles in Methicillin-Sensitive S. aureus Pathotypes Isolated in a Commercial Rabbit Farm in Italy.”
The paper is of high value due to its original character; it treats a specific subject that is of high interest for the domain of food microbiology, food safety, for the food chain, and the public health. With some minor exceptions (which refers to some descriptions necessary), all materials and methods are specified and described adequately.
All iconographic materials – five tables and six figures - were given accurate descriptions, the results were described in great detail, and the conclusions are adequate.
Antimicrobial resistance (AMR), defined as the ability of microorganisms to resist antimicrobial treatments, especially antibiotics – has a direct impact on human and animal health and carries a substantial economic burden due to higher costs of medications and reduced productivity caused by sickness. At the level of the European Union, for example, AMR is responsible for an estimated 33,000 deaths per year. Supplementary, it is also expected that AMR costs the European Union 1.5 billion EUR per year in healthcare costs and productivity losses.
The present paper approaches an interesting topic, that is, the prevalence of Methicillin-Sensitive Staphylococcus aureus (MSSA), and secondary, the description of the main characteristics of Methicillin-Sensitive Staphylococcus aureus (MSSA) – all these pathotypes of staphylococci isolated from animals of different ages, and its handlers (farm-workers). Even though the study does display certain limitations, the approach to the topic itself is a solid one, well-argued and unequivocal.
With some minor exceptions, the paper is well structured and possesses a high novelty character. The major components of the article – Introduction; Results; Discussion, and Materials and Methods - are organized judiciously and in direct connection one with another. I believe that in connection with the organization of the paper, the following clarifications are required: the chapter entitled materials and methods usually follow immediately after the introduction: also, it is necessary to introduce a separate final chapter, reserved for conclusions.
The documentation is adequate, and all the authors are cited in the text of the paper. The authors of the article need to pay more attention to writing (editing of text): the existence of some small writing errors (errors of editing) makes it harder to check the citations (checking the authors from the bibliographic reference list), and it can create some confusion in terms of understanding specialized terms.
The provided scientific results are exact and precise. The goal of the conducted research is well specified and delineated. The working protocol is appropriate, and the used analysis methods are coherent with the proposed objectives.
Nevertheless, the detailed analysis of the paper has also highlighted some aspects that require revision, as follows below:
The bibliography is relevant but presents some minor lacks when it comes to citations and mentions. To clarify some aspects, I would suggest that the authors write the references list evenly: for example, journal papers require either the complete journal name, or the JCR abbreviation (in the case of ISI indexed or rated journals), or the ISO abbreviation (for BDI indexed journals); moreover, for journals, I suggest that the volume, number, and pages (as the case requires) be mentioned.
For example – page 13, lines 412-413, number 2 in the bibliographic references list - Tyrrell K.L., Citron D.M., Jenkins J.R., Goldstein E.J.C. Periodontal Bacteria in Rabbit Mandibular and Maxillary Abscesses. Journal of Clinical Microbiology (or JCR Abbreviation – J. Clin. Microbiol.), 2002, 40, 3, 1044-1047; DOI: 10.1128/JCM.40.3.1044–1047.2002.
Another example – page 13, lines 424-426, number 8 in the bibliographic references list - Agnoletti F., Mazzolini E., Bacchin C., Bano L., Berto G., Rigoli R., Muffato G., Coato P., Tonon E., Drigo I. First reporting of methicillin-resistant Staphylococcus aureus (MRSA) ST398 in an industrial rabbit holding and in farm-related people. Veterinary Microbiology (or JCR Abbreviation – Vet. Microbiol.), 2014, 170, 1-2, 172-177; DOI: https://doi.org/10.1016/j.vetmic.2014.01.035.
Under these circumstances, the additional mention of the Digital Object Identifier (DOI) becomes optional.
The observation is valid for all the articles from the bibliographic references list that are incompletely formulated.
I would also recommend that greater attention be paid when it comes to chapters from books and that the number of pages, the publishing houses, and other identification elements (link, Digital Object Identifier – DOI, etc.) be mentioned, regardless of the reference type.
The mentioning of the authors in the list of references in alphabetical order, from A to Z, is also recommended: thus, the text becomes way more readable, and the cited authors are more visible and easy to find and verified. This is important because, generally, there may be authors with works from different years.
Moreover, I suggest that the authors consult and include the following papers in the list of references:
Bondoc I. European Regulation in the Veterinary Sanitary and Food Safety Area, a Component of the European Policies on the Safety of Food Products and the Protection of Consumer Interests: A 2007 Retrospective. Part One: the Role of European Institutions in Laying Down and Passing Laws Specific to the Veterinary Sanitary and Food Safety Area. Universul Juridic, Supliment, 2016, pp. 12-15 (Available online: http://revista.universuljuridic.ro/supliment/european-regulation-veterinary-sanitary-food-safety-area-component-european-policies-safety-food-products-protection-consumer-interests-2007-retrospective/).
Bondoc I. European Regulation in the Veterinary Sanitary and Food Safety Area, a Component of the European Policies on the Safety of Food Products and the Protection of Consumer Interests: A 2007 Retrospective. Part Two: Regulations. Universul Juridic, Supliment, 2016, pp. 16-19 (Available online: http://revista.universuljuridic.ro/supliment/european-regulation-veterinary-sanitary-food-safety-area-component-european-policies-safety-food-products-protection-consumer-interests-2007-retrospective-2/).
All these papers approach the matter of food safety legislation enforced within the European Union, which usually constitutes a blueprint for the law in third countries. The two documents outline the European legislative environment, starting with the year 2007, the year of the penultimate geo-political enlargement of the European Union. I want to add that all recommended papers have been indexed in CAB International and HeinOnline, the largest and most extensive worldwide database for documents in the legal field. They are also indexed in other databases: Google Scholar, ResearchGate, Web of Science, Central and Eastern European Online Library (CEEOL), and others.
The authors should pay more attention to the use of certain abbreviations to avoid confusion; basically, all abbreviations are to be used in the text-only after at least one mention made in extenso.
The obtained results are interpreted correctly, and their practical value is visible: however, some data are not clearly expressed in the text, situations that need to be remedied by the authors.
The graphical representation of the results is adequate: both tables and figures have appropriate titles for the presented data. The submitted data are factually correct, with a logical and coherent connection between the data in the text and those in figures and tables. However, a clarification is required: table 5 is present only in the supplementary material, although it is referred to in the text of the article (page 9).
As for the grammar of the paper, most of the text is very well written, with few parts that would require some modifications – just a few small recommendations, as follows:
Page 1, line 30 – replace “The number of resistances” with “The resistance”;
Page 1, line 32 – replace “a higher number of resistances” with “a higher resistance”;
Page 2, line 71 – replace “As reported previously” with “As it was previously reported”;
Page 4, line 111 – replace “to age” with “to the age”;
Page 5, line 132 – replace “resistances” with “resistance”;
Page 6, line 166 – replace “Out of 7 rabbits” with “Out of the 7 rabbits”;
Page 7, line 199 - replace “The number of resistances” with “The resistance”;
Page 8, line 203 – replace “the highest average number of resistances” with “the highest average resistance value”;
Page 8, line 207 – modify ”According to the age categories, the resistance (...).’’
Page 8, line 212 – replace “in farm” with “in the farm”;
Page 8, line 213 – replace “the number of resistances” with “the resistance types”;
Page 8, line 216 – replace “the number of resistances” with “the resistance types”;
Page 9, line 221 – replace “in farm” with “in the farm”;
Page 9, line 234 – replace “resistances” with “resistance”;
Page 10, line 259 – replace “portion” with “group”;
Page 10, line 260 – replace “very low” with “very reduced”;
Page 11, line 296 – replace “resistances can hide completely” with “resistance can completely hide”;
Page 11, line 304 – replace “of sanitary” with “of the sanitary”;
Page 11, line 337 – replace “for representation of” with “for the representation of”;
Page 12, line 360 – replace “that lived in farm” with “that have lived in the farm”;
Page 12, line 379 – replace “and association” with “and the association”;
Page 13, line 394 – replace “database” with “a database”.
Minor corrections and clarifications notwithstanding, the authors’ work and obtained results are highly commendable. They bring significant added value to the work and may constitute a launching pad for further useful studies.
Provided that the authors verify the paper and perform the required corrections, the article may be accepted and published in the Antibiotics.
Best Regards,
Reviewer
Author Response
ANTIBIOTICS - 901983, REVISION NOTE
Reply to the comments on manuscript “Analysis of the Antibiotic Resistance Profiles in Methicillin-Sensitive S. aureus Pathotypes Isolated on a Commercial Rabbit Farm in Italy “
Author’s Response to the Reviewer’ Comments
Reviewer #3
The authors thank the reviewers for their useful comments and suggestions to the manuscript.
All comments have been taken in due consideration and the corrections added to the original paper. In the manuscript all corrections are highlighted in yellow colour.
The present version of the manuscript has been checked by the MDPI’s English editing service.
We hope that the manuscript is now suitable for publication.
Thank you again for your consideration
Sincerely,
The authors
I believe that in connection with the organization of the paper, the following clarifications are required: the chapter entitled materials and methods usually follow immediately after the introduction.
We agree with you, but we followed the Journal’s instructions. We have inserted each chapter in the order requested (Introduction, Results, Discussion, Materials and Methods).
…also, it is necessary to introduce a separate final chapter, reserved for conclusions.
Following the suggestion, we have added the chapter “Conclusions” (lines: 331-344).
The authors of the article need to pay more attention to writing (editing of text): the existence of some small writing errors (errors of editing) makes it harder to check the citations (checking the authors from the bibliographic reference list), and it can create some confusion in terms of understanding specialized terms.
We apologize for editing errors. We have reviewed the text corrections and the manuscript was further checked by the MDPI’s English editing service. In the manuscript all corrections are highlighted in yellow colour.
…Nevertheless, the detailed analysis of the paper has also highlighted some aspects that require revision, as follows below: The bibliography is relevant but presents some minor lacks when it comes to citations and mentions. To clarify some aspects, I would suggest that the authors write the references list evenly: for example, journal papers require either the complete journal name, or the JCR abbreviation (in the case of ISI indexed or rated journals), or the ISO abbreviation (for BDI indexed journals); moreover, for journals, I suggest that the volume, number, and pages (as the case requires) be mentioned. Under these circumstances, the additional mention of the Digital Object Identifier (DOI) becomes optional.
We have verified that we had followed the guidelines of the journal.
We have added the DOIs, as you proposed.
The observation is valid for all the articles from the bibliographic references list that are incompletely formulated. I would also recommend that greater attention be paid when it comes to chapters from books and that the number of pages, the publishing houses, and other identification elements (link, Digital Object Identifier – DOI, etc.) be mentioned, regardless of the reference type.
The details about the references were added and now they are in agreement with the journal’s instructions.
The mentioning of the authors in the list of references in alphabetical order, from A to Z, is also recommended: thus, the text becomes way more readable, and the cited authors are more visible and easy to find and verified. This is important because, generally, there may be authors with works from different years.
We agree with you, but we followed the journal’s instructions, (Antibiotics’ guidelines), therefore the list of references is in the order in which it is shown in the text.
Moreover, I suggest that the authors consult and include the following papers in the list of references: Bondoc I. European Regulation in the Veterinary Sanitary and Food Safety Area, a Component of the European Policies on the Safety of Food Products and the Protection of Consumer Interests: A 2007 Retrospective. Part One: the Role of European Institutions in Laying Down and Passing Laws Specific to the Veterinary Sanitary and Food Safety Area. Universal Juridic, Supplement, 2016, pp. 12-15 Bondoc I. European Regulation in the Veterinary Sanitary and Food Safety Area, a Component of the European Policies on the Safety of Food Products and the Protection of Consumer Interests: A 2007 Retrospective. Part Two: Regulations. Universal Juridic, Supplement, 2016, pp. 16-19.
Thank you for this suggestion, it would have been interesting to include the European regulation of the veterinary sanitary. However, our study is focused only on microbiological aspects (antibiotic resistance in S. aureus) and we believe that regulations are out of context in the manuscript.
The authors should pay more attention to the use of certain abbreviations to avoid confusion; basically, all abbreviations are to be used in the text-only after at least one mention made in extenso.
We apologize, but we have been misled by the inclusion in the back of the manuscript of the ‘materials and methods’ chapter. We have checked that abbreviations are used in the text only after mention in extenso.
The obtained results are interpreted correctly, and their practical value is visible: however, some data are not clearly expressed in the text, situations that need to be remedied by the authors..
The authors tried to better clarify (lines 107-109, 114-115). The authors have reviewed the text corrections and the manuscript was further checked by the MDPI’s English editing service. In the manuscript all corrections are highlighted in yellow colour.
…, a clarification is required: table 5 is present only in the supplementary material, although it is referred to in the text of the article (page 9).
The authors should have solved the problem of the disappearance of table 5. It was in the manuscript but hidden. It is now correctly visible. In case, it appears by clicking on the triangular icon.
Page 1, line 30 – replace “The number of resistances” with “The resistance”. OK, it has been done.
Page 1, line 32 – replace “a higher number of resistances” with “a higher resistance. OK, done.
Page 2, line 71 – replace “As reported previously” with “As it was previously reported”. OK, done
Page 4, line 111 – replace “to age” with “to the age”. OK, done.
Page 5, line 132 – replace “resistances” with “resistance”. OK, done.
Page 6, line 166 – replace “Out of 7 rabbits” with “Out of the 7 rabbits”. OK, done.
Page 7, line 199 - replace “The number of resistances” with “The resistance. OK, done.
Page 8, line 203 – replace “the highest average number of resistances” with “the highest average resistance value”. OK, it has been done.
Page 8, line 207 – modify ”According to the age categories, the resistance (...).’’ OK, done.
Page 8, line 212 – replace “in farm” with “in the farm”. OK, done.
Page 8, line 213 – replace “the number of resistances” with “the resistance types”. OK, done.
Page 8, line 216 – replace “the number of resistances” with “the resistance types”. OK, done.
Page 9, line 221 – replace “in farm” with “in the farm”. OK, done
Page 9, line 234 – replace “resistances” with “resistance”. OK, done.
Page 10, line 259 – replace “portion” with “group”. OK, done.
Page 10, line 260 – replace “very low” with “very reduced”. OK, done.
Page 11, line 296 – replace “resistances can hide completely” with “resistance can completely hide”. OK, done.
Page 11, line 304 – replace “of sanitary” with “of the sanitary”. OK, done.
Page 11, line 337 – replace “for representation of” with “for the representation of”. OK, done.
Page 12, line 360 – replace “that lived in farm” with “that have lived in the farm”. OK, done.
Page 12, line 379 – replace “and association” with “and the association”. OK, done.
Page 13, line 394 – replace “database” with “a database”. OK, done.
Round 2
Reviewer 1 Report
The paper has been improved in relation to the comments. However, that conclusion should be more specific; they are now too general.